# The Halo Effect of CSR Activity: Types of CSR Activity and Negative Information Effects

**Chang-Hyun Jin [1],\* and Jung-Yong Lee [2]**

[1]   Department of Business Administration, Kyonggi University, Gyeonggi-do 16227, Korea
[2]   Department of Business Administration, SungKyunKwan University, Seoul 03063, Korea;
      jaylee8206@gmail.com
\*    Correspondence: chjin@kgu.ac.kr; Tel.: +82-31-249-9427

**Abstract:** This study attempts to assess the role of the polarity of information or publicity about a company (whether positive, negative, or neutral) and two CSR activities (e.g., environmental preservation and supporting social welfare in developing countries) in the relationships between consumers and brands and also to investigate how a corporate image, as perceived by consumers, affects the formation of an image of a company or brand through the halo effect of Corporate Social Responsibility (CSR) activity. An experimental design was used to test the hypotheses. A group of subjects who were exposed to negative publicity about a company showed a change in average values in their attitudes toward the brand, purchase intention, corporate image, attitude to the CEO, and brand loyalty. The formation of attitudes or images changed more positively when the type of CSR activity involved was closely related to a company's corporate image. The results of the analysis used to test the halo effect of a company's CSR activity indicate that corporate CSR activity is closely related to consumer attitude formation or changes in perceptions of a company. The results of this study provide an opportunity to assess the importance of negative information about a company or product, as well as types of CSR activity that affect image formation. The study suggests that only CSR activities that are highly congruent with a company's image or its products can produce positive and amicable reactions from consumers through the halo effect.

**Keywords:** consumer attitude; CSR activity; halo effect; negative effects

## 1. Introduction

Companies seek business performance success by establishing corporate images that suggest that they are sensitive to the interests of stakeholders, local residents, and organizational members, as well as enacting social values that contribute value to society. The responsibility for enhancing human value, beyond creating economic value, based on civic social awareness is regarded by some as a condition for corporate success. In this context, corporations have embraced Corporate Social Responsivity (CSR) to address industrial, societal, and cultural expectations [1].

In a business environment marked by increasingly fierce market competition, building a strategy that combines differentiated CSR tactics and specific execution plans aimed at improving a corporate image and management performance is becoming more and more difficult. The increasingly diversified needs of consumers and various stakeholders also make it difficult to establish an effective CSR strategy [2,3]. Still, studies have shown that a company's CSR activity can improve both brand valuation and corporate image [4]. Moreover, the intentions, motivations, differentiation, sustainability, and friendliness of a given CSR activity must be considered if a company hopes to translate the adoption of that activity into business success.

CSR should provide a company with an effective marketing communication tool with which to burnish its corporate image in the public's mind [5–8]. A multinational company must establish a strong brand to build strong relationships with consumers. CSR activities, if properly communicated to the public, should contribute to such a positive brand image. Thus, effective brand management requires an understanding of how CSR activities affect consumer attitude formation through their effects on brand image.

CSR research remains underdeveloped. Some researchers have raised concerns because research findings regarding the effects of CSR activity have been mixed, and in some cases controversial, primarily because there are no consensus on measures of CSR performance, leading some to argue that studies of the effects of CSR activity lack validity or credibility [9]. We designed the present study to enhance the credibility of measures of CSR activity by examining the effects of negative information or publicity and applying the concept of the halo effect.

The main purpose that the study attempts to explore whether there is an interaction effect between exposure to polarized corporate information and two CSR activities. How consumer attitudes towards a company differ after exposure to polarized information about the company (negative publicity and descriptions of the company's CSR activity). The study also examines the halo effect of CSR activity play when consumers are exposed to polarized information about a company.

Thus, the purpose of this study is (1) to access the role of the polarity of corporate information (whether positive, negative, or neutral) and two CSR activities (environmental preservation and social welfare support in developing countries) on consumer attitudes, intention to purchase, and other brand-related factors; (2) to provide evidence pertaining to the roles that negative information or publicity and CSR activity play in affecting brand evaluation, the formation of a corporate image, consumer attitudes toward a CEO, and brand loyalty building; (3) to investigate how consumer perceptions of a corporate image affects the formation of that corporate image and the corresponding brand through the halo effect; and (4) to examine how consumer attitudes change before and after exposure to corporate information or publicity and information about CSR activity.

## 2. Theoretical Background

### 2.1. Corporate Social Responsibility

CSR has attracted scholarly attention in management and marketing studies. A company's commitment to sustainability and social responsibility lies at the core of the new paradigm of the global corporate management environment [10–12]. CSR has evolved such that not only large companies, but also small and medium-sized companies must consider adopting it as a strategy [1,13]. Some researchers have even argued that CSR is an essential strategy [9]. Others argue that the public can feel a sense of kinship with companies that implement CSR, which increases interest in those companies [14].

Although CSR activity has been found to positively affect a company's marketing and management performance [5–8], such activity also increases costs [7]. Moreover, CSR activity can have a negative effect when consumers perceive that a company's motivation for pursuing that activity is not pure [2] or that the company has weak innovative power [15]. That is, consumers sometimes doubt the authenticity of a firm's publicly stated commitment to CSR [2], as they expect companies to be sincerely committed to CSR, not to adopt it simply try to look good to increase returns or profits [16,17]. It is difficult to evaluate the social value of CSR. CSR activity often fails to achieve the intended social change. Some studies have even reported that companies that fulfill their social responsibility exhibit poor financial performance due to opportunity loss and increased costs [18,19].

On the positive side, CSR activity can act as a buffer that reduces the impact of an event that can lead consumers to form a negative image of a company [13,19]. CSR can also help a company build long-term relationships that help it minimize negative industry impacts [1,20]. Corporate investment in CSR activity can therefore help a firm respond positively to stakeholder concerns and improve

social conditions, enhancing its reputation [6,7,13,21–23]. Ultimately, CSR has a significant impact on the formation of a positive corporate public image [5–8]. More broadly, CSR can be seen as a way of forming intangible assets, providing insurance that diminishes the negative effects of an adverse event on a company's management performance [13,24]. Godfrey and Vanhamme and Grobben also find that having an image or reputation as a good company due to CSR activity can diminish the negative impact of a negative event [24,25].

### 2.2. Negative Information Effects

Negative information or publicity can harm the reputation or perceived value of people, products, issues, or a company. Once people are exposed to a negative message, it is stored in both long-term and short-term memory and affects current or future decision-making processes [26]. Damage caused by a product, food poisoning caused by a certain brand, unethical treatment of workers, or inappropriate business practices can all generate negative information about a company or the reliability of its products [27,28]. In general, negative information is revealed by mass media, reports from civic groups, or dissatisfied consumers' complaints [13,28,29].

Exposure to negative information about a company or product affects the company's reputation and the intention to purchase the product, ultimately reflecting poorly on corporate management [25,28]. Klein and Dawar and Van Heerde et al. find that negative information not only damages a company's reputation, but also reduces market share and brand equity [13,30]. Compared with positive information, negative information more easily attracts consumer attention and occupies a greater share of reported information. Negative information sometimes circulates in public discourse through rumors [31].

In the absence of a direct relationship between a company's social activities and its products, compensatory inferences may negatively affect purchase behavior and attitude formation [15,31]. On the other hand, based on the above discussion, social expectations associated with CSR give it a positive impact on consumer perceptions of a company, its CEO, and its brand, positively affecting purchase behavior. This brings us to the halo effect.

### 2.3. The Halo Effect

The halo effect was first discovered by Well, who posited a phenomenon whereby general opinion about a certain object affects the evaluation of specific characteristics of the object [32]. Newcomb defined the halo effect as a logical error that occurs when a person evaluates logically unrelated behaviors similarly [33]. Other scholars have defined the halo effect as a tendency for a consumer's beliefs about a brand name or its reputation to affect purchase behavior [14,34,35].

The overall image of a product or company is formalized by several components. An image affects the evaluation of other related characteristics once it is formed. In general, this phenomenon is connected to the halo effect, which means that a comprehensive image of a certain object affects the evaluation of other characteristics of the object. Researchers have investigated the halo effect from a variety of perspectives. They have commonly argued, however, that consumers make evaluations based on comprehensive impressions, rather than evaluating a product based on specific aspects of its performance.

In a study examining the halo effect of an image in relation to a manufacturing company, Schiffman explained that a company's image induces consumers to draw a certain conclusion before exploring other information about a product when they have very little experience with or information about the product [36,37]. Chernev and Blair mentioned that a company's CSR activity can give rise to a halo effect that will influence consumer perceptions of the company's reputation [15]. The general impression of the evaluated object or the evaluation of its prominent characteristics becomes the origin of the halo [38]. Previous research has pointed out that a company's marketing communication strategy, which can include CSR, generates a halo effect through consumer awareness of, and goodwill

for, the perception of an activity that is favored by consumers [14,39]. In contrast, any resulting halo effect is an intangible construct, creating a measurement challenge [14,39].

## 3. Hypotheses and Theoretical Model

### 3.1. Negative Information and Types of CSR Activity

This study explores the role of the polarity of corporate information (whether positive, negative, or neutral) in consumer evaluation of a company's brand and image by focusing on subjects' exposure to negative information about a company and its CSR activity in the context of evaluating the company and its CEO as well as eliciting subjects' expressions of brand attitude and brand loyalty.

Negative information about a company revealed by mass media not only undermines its credibility in consumers' minds but can also create a crisis for the company [40–45]. Additionally, it has been found that negative information attracts consumer attention more compellingly than positive information [28,46]. Consumers who are exposed to negative information about a company or its products will likely evaluate the company negatively [13,26,29,31]. Consumer perceptions of negative information can change depending on the intention behind, the accountability for, and the intensity of the negative information.

The time dependence of evaluations, attitudes, and intentions is a central factor in the consumption-systems approach, according to which a company's offerings are refined as a bundle of goods and services that are consumed over time, in multiple consumption episodes [47,48]. Some scholars have found that behavioral intentions over a given period of time are a function of attitudes formed during that period and intentions formed in prior periods [47,48]. We conducted an experiment that measures responses to stimuli, at two times, to polarized information about a company (negative publicity about a company paired with positive information about its CSR activities) to measure the effects on consumer attitude formation before and after exposure to these stimuli.

**H1:** *When subjects are presented with negative information about a company, attitude towards the brand (H1-1), purchase intention (H1-2), corporate image (H1-3), attitudes towards the CEO (H1-4), and brand loyalty (H1-5) will be more negative than they were before exposure to the negative information.*

**H2:** *When subjects are presented with positive information about a company, attitudes toward the brand (H1-1), purchase intention (H1-2), corporate image (H1-3), attitudes towards the CEO (H1-4), and brand loyalty (H1-5) will be more positive than they were before exposure to the positive information.*

Previous researchers have found that consumers exhibit favorable responses to CSR activities [13,15,21]. Many studies have analyzed the direct effects of CSR on consumer attitude formation, purchase behavior, and loyalty [49–51]. Even if CSR activity does not instantly produce favorable outcomes for companies, it ultimately plays an important role in reducing damage to a company's reputation in a crisis situation by affecting consumers' attribution behavior [13,52,53]. When a negative message about a certain company becomes public, having a corporate image that has been established through CSR activity acts as a cushion against negative evaluations. This suggests the strategic wisdom of engaging in CSR. Becker-Olsen et al. find that the effects of CSR activity differ according to the congruence of the subject of CSR activity with a company's characteristics [2].

CSR activity influences the introduction stage of the purchase process by enhancing cognitive stimuli when consumption is driven by pro-social activities [16,20,21]. CSR activity has a positive impact on consumer attitudes towards a product and on the intention to purchase that product [2,9,12,53,54].

Menon and Kahn and Rifon et al. relate the effects of CSR activity to factors such as whether a company or brand image obligates the company to adopt CSR activity [55,56]. CSR activity is more effective when it is combined with a positive corporate or brand image. That consumers have exhibited amicable attitudes towards companies that carry out differentiated public activities indicates that the effects of CSR activity are stronger on consumers who believe that CSR is important [57,58].

A company's CSR activity attracts the greatest interest when it involves local issues, followed by domestic issues and global issues [59]. Thus, this study assumed that the types of pro-social activities in which a company engages will influence consumer attitudes toward the company.

**H3:** *When exposed to information about a company's CSR activity that focuses on environmental preservation, attitude towards the brand (H3-1), purchase intention (H3-2), corporate image (H3-3), attitude towards the CEO (H3-4), and brand loyalty (H3-5) are more positive than before exposure to the company's environmental preservation efforts.*

**H4:** *When exposed to information about a company's CSR activity that focuses on supporting social welfare in developing countries, attitude toward the brand (H4-1), purchase intention (H4-2), corporate image (H4-3), attitude towards the CEO (H4-4), and brand loyalty (H4-5) are more positive than they were before exposure to the company's support of social welfare in developing countries.*

*3.2. Research Questions*

Studies have suggested that a company seeking to establish an identity in a foreign market should develop an image as a good corporate citizen by making CSR activities a routine part of its business processes [60–62]. CSR activity has considerable influence on consumer attitudes towards and intention to purchase a product [63]. Brown and Dacin argue that CSR activity significantly affects how consumers evaluate a company or product and report an association with CSR activity that induces consumers to believe that a company manufactures high-quality products and that a company that pursues CSR activities receives positive consumer evaluations of a new product [63].

Klein and Dawar argue that a halo effect occurs when a company is associated with CSR activity, a finding based, in part, on three concepts that are central to attribution theory—accountability, stability, and controllability, and find that these factors are connected to brand evaluation by consumers [64,65]. CSR activity does not produce instant benefits for companies, but it plays an important role in mitigating potential harm in brand evaluation when a crisis situation that affects consumer attribution behavior occurs. Moreover, CSR activity has a positive impact on the intention to pay premium prices for a product and the intention to invest or work in a company [64,66]. This study will examine whether there is an interaction effect between exposure to polarized corporate information and two CSR activities. Thus, the study seeks to answer the following research question:

RQ1: What role does the halo effect of CSR activity play when consumers are exposed to polarized information about a company?

**4. Research Method**

This study examines whether a company's CSR activity acts as a buffer that reduces the impact of negative events. We investigate how such information induces consumers to rethink or change their images of a company or brand by distinguishing between consumer attitudes before exposure to information about a company and after such exposure. We applied the concept of the halo effect to examine whether CSR activity itself is involved in the formation of company or brand attitudes and whether consumer exposure to CSR activity changes such perceptions.

The present study employed a 2 × 2 × 2 factorial between-subject design (exposure (before vs. after) × polarity information (negative vs. positive information) × 2 CSR activities (relevant CSR activity vs. non-relevant CSR activity) was used for this study. Attitude towards the brand, purchase intention, corporate image, attitude towards the CEO, and brand loyalty were treated as independent variables in this study. The dependent variable is consumer intention to take action in light of the negative information about the company, with reference to its highly relevant social activities.

### 4.1. Pretest and Experimental Stimulus Selection

The study conducted a pretest to check not only the experimental design but also the selected experimental stimulus. Experimental materials were selected by the researcher and two experts who are specialized in marketing. A global mobile communication company was chosen as the experimental object. The target company is a major mobile device seller, so participants were familiar with both the company and its product. Study participants should therefore find it easy to evaluate the company's brand and CEO. Research conducted to measure brand loyalty to a real company or brand should not, however, include consumers with strong loyalty to the company or brand [58]. Whether participants in this study brought high or low levels of loyalty to the company into the experiment, there is little concern about sample bias because the study focuses on the relationship between the effects of negative information about the company's management and the effects of positive information about the company's CSR activity. We discuss the issue of natural and artificial stimuli in the experimental design when we acknowledge the study's limitations.

Experts (e.g., professors, marketers, and doctoral students) in the relevant field recommended appropriate stimuli for the study. In particular, they recommended specific CSR activities and negative information as experimental stimuli. Reflection on the purpose of the study was considered when choosing the experimental stimuli. The negative and positive information presented in the experiment pertained to the experimental object, which consisted of a company, its products, and its CEO. For negative information, the author edited information from a document that describes the company's alleged accounting fraud and tax evasion and its chief executive's external activity and response, and standardized the form of the document. For positive information, information related to the company's CSR activity, new investments, the chief executive's volunteer activity, its strategy for new technology development, charitable giving, and awards for service quality were compiled.

To choose the CSR type that is related to a corporate product or image, this study first chose a type of CSR that reflected the image of the company and then chose another type that is less associated with that image. The study thereby chose two types of CSR activities and edited the contents of those activities based on information about the company. The first type of CSR activity is related to preventing environmental pollution. Specifically, the activity involves the collection of used mobile phones, which corresponds to the image of a mobile phone company and its products. The second CSR activity involves global social welfare, an educational project that this company executes and uses to promote the improvement of social welfare in developing countries.

A total of 68 undergraduate students in a department of business administration at a larger university in Korea, who had taken a marketing course, participated in the pretest. Investigators asked the experimental subjects to read information about the company that was polarized, with the information representing the negative and positive poles randomly assigned to the subjects. All subjects were undergraduate business administration students who received extra course credit for participating in the study. The purpose of the pretest was to check the experimental procedure, the composition of the questionnaire, the editorial condition of the document, and its information content, before running the actual experiment. A five-point Likert scale was used to measure the negative and positive information. To test the separated or polarized information in this study operationally, we referred to a previous study [67]. A question asking for subjects' opinions after reading a piece of information was measured against a 5-point index, where 1 indicates "very negative" and 5 indicates "very positive." T-test results indicated the significance of the operational test of the experimental stimuli. The average value of the negative environmental information was 2.56 (S.D. = 0.040) and the average value of the positive information was 3.17 (S.D. = 0.099), both of which were statistically significant ($t = 6.73$, $p < 0.001$).

To test the halo effect according to type of CSR activity, the study chose two types of CSR campaigns. The study first chose a total of five types of CSR activities that were suggested by a previous study, which included social welfare, education, culture and art, environmental preservation, and global social welfare support projects, and let students who participated in the pretest rank the

five types. The participants determined the rank by giving the highest score to the CSR type that is most closely related to the mobile communications company that was the focus of the experiment. Subjects simultaneously chose the type of CSR activity with a strong match to the company and that with a weak match with the company. The study conducted experimental stimulus tests using the type that had the highest score and the type with the lowest score. The average was highest in the case of environmental preservation (M: 3.29) and the lowest was in the case of global social welfare support (M: 2.01).

As noted above, the CSR activity with an environmental preservation orientation involved a campaign of collecting used mobile phones, which fits the mobile communications company. The information on this campaign explained that the relevant company is the leader of an environmental protection campaign and the rewards for customers who bring their used mobile phones include loyalty point savings, gift cards, and trade-in offers.

The second CSR activity, as noted above, is a global support project that involves medical support for developing countries. The material presenting the CSR activity, consisting of the social welfare project, explains that the company is participating in providing medical support and initiation of greater medical capacities in developing countries. To test the second CSR activity operationally, we referred to a previous study [67]. Test questions asked whether the social responsibility activities that this company is executing are related to the company's corporate image, purpose, and products. A total of three question items were measured on a 5-point scale. The average for the CSR activity consisting of environmental preservation was 3.29 (S.D. = 0.033) and the average for the global support CSR activity was 2.01 (S.D. = 0.068), both of which were statistically significant ($t = 18.57$, $p < 0.001$).

Insofar as this study designed an experiment using an actual company as an object, to ensure the equivalency of the experimental stimuli between the groups, we implemented a pretest of brand loyalty and excluded subjects who showed extreme brand loyalty from the sample. Since loyalty to a company can affect experimental results, the study removed this exogenous variable. All three of the question items regarding brand loyalty were tested on a 5-point index. Experimental subjects returning the lowest score of 1 or the highest score of 5 were excluded.

### 4.2. Experimental Procedure

Two hundred undergraduate students in two marketing classes from the same university in Korea were recruited for the experiment. As noted, the subjects were awarded extra course credit. They had extensive knowledge about CSR and corporate ethics based on courses they had taken, such as business ethics and management in society. A total of 51% of the participants were male and 49% were female. To prevent problems, such as leakage of information about the experimental stimuli prior to the launch of the experiment, we implemented the process in separate places at the same time. Subjects had no previous exposure to either experimental stimulus. A random sampling method in SPSS was used for group allocation.

The first experiment examined how consumers' attitudes changed based on the type of CSR activity involved, as well as based on exposure to information related to the company. The experiment was conducted over a period of one month. In the first week, the researcher asked participants to write about the image of the company and brand and express their attitudes towards it. The company logo and image were placed at the top of the questionnaire and questions were arranged in the following order: Attitude towards the brand, purchase intention, corporate image, attitude towards the CEO, and brand loyalty. For the purpose of investigating changes in attitude formation depending on both information type and CSR type, the first experiment divided the sample into four groups, as follows: Two groups according to types of information and two groups according to types of CSR activity.

In the second week of the experiment, experimental stimuli containing negative and positive information about the company and information on the two types of CSR activity it pursues was edited by the researcher and distributed to the experimental subjects, who read the material separately. Considering the intellectual level of an ordinary adult, the reading material was presented on a

half-page of A4 paper so that it could be read within three minutes. In the final and third week, participants read the information in an identical way. After reading the material within a given time period, subjects wrote answers to the questionnaire regarding their attitudes towards the company and brand.

The second experiment focused on the halo effect. The experiment was designed to investigate differences in attitudes before and after exposure to the stimuli that were identical to those used in the first experiment, which included two types of information and two types of CSR activity. Two hundred and eighty undergraduate students were recruited from the same university. The sample was divided into four groups. Subjects in the experiment responded to the survey regarding the company and brand under the same conditions before exposure to the stimuli. Subjects were allocated randomly to each group. The experiment proceeded based on a between-subject factorial design using a total of four groups (Types of information: Negative/positive. Types of CSR activity: Environmental preservation/global support). The study investigated changes in attitudes in each group before and after exposure to the stimuli.

The experiment was conducted over a period of one month. In the first week, participants evaluated the company and brand. In the second week, subjects read the stimuli consisting of edited material for 10 minutes. In the final week, subjects were exposed to the stimuli in the same way. They then wrote answers to the questionnaire regarding the company and brand after a set period of time. In our study, exposure to the stimuli occurred twice, following the method of previous studies involving exposure to experimental stimuli.

### 4.3. Measurement

To measure attitudes toward the brand, we applied three items developed by MacInnis and Park [68]. Questions involved level of feeling favorable, level of loving, and level of liking and were measured using five-point Likert scales.

To measure intention to purchase, we used three items from the measurement index suggested by Haley and Case [69]. Intention to purchase was measured using a single index that asks about the intention to buy a product from a target company when respondents must buy a similar product in the foreseeable future. The questions included "I am interested in buying this product", "I am very likely to buy this product," and "I would generally like to buy products from this company".

For questions related to corporate image, we applied three items that were used in Sen and Bhattacharya [70] and Rio et al. [71]. Specific question items included level of good feeling, level of favorable feeling, and level of credible feeling. To measure attitudes toward the CEO, we recomposed questions used in Jin and Yeo [66] to fit this study. These questions included "I like the CEO of this company", "I have a favorable impression of the CEO of this company", and "I think the CEO of this company is trustworthy".

Brand loyalty reflects consumer attachment to or sentiment about a certain brand [71]. To measure brand loyalty, the study recomposed question items suggested by Chaudhuri and Holbrook and created three question items that fit this study [72], which included level of recommendation to other people, trust, and level of continuous use. For example, "I am willing to recommend the brand to others."

### 4.4. Manipulation Check

We used three items to check the validity of the experimental stimuli, which included two types of information and two types of CSR activity. First, we conducted t-tests to check the significance of the mean of each group. The mean value of subjects' responses in each experimental group was significant. The results can be summarized as follows: For the operational test of the negative information type ("This information is very negative"/"This information is not negative at all"), the results were $3.59/1.54$, $t = 32.5$, $p < 0.001$; for the operational test of the positive information type ("This information is very positive"/"This information is not positive at all"), the results were $3.59/1.54$, $t = 29.5$, $p < 0.001$;

for the operational test of the environmental preservation CSR activity, which used the CSR activity and the company and brand images (similar/dissimilar), the results were 3.67/2.20, $t = 34.7$, $p < 0.001$; for the operational test of the global social welfare support CSR activity, which also used image congruence (similar/dissimilar), the results were 2.31/3.53, $t = -18.7$, $p < 0.001$. The reliability of the index was evaluated using Cronbach's alpha. The Cronbach's alpha coefficient came in between 0.892 and 0.94, satisfying the criterion, confirming the reliability of the measurement tool.

## 5. Results

### 5.1. Testing the Hypotheses

To test the effectiveness of the sub-factors comprising each variable, we conducted multivariate analysis of covariance (MANCOVA) using pre-test scores as the covariate. A MANCOVA tests differences in the values of the dependent variable.

As seen in Table 1, analysis of the differences in the dependent variables before exposure and after exposure produced Wilks's $\lambda = 0.928$ ($F = 3.496$, $p = 0.002$). In the case of analyzing differences between the negative and positive information groups, Wilks's $\lambda = 0.951$ ($F = 2.329$, $p = 0.033$). In the case of analyzing between-group differences according to types of CSR activity, Wilks's $\lambda = 0.864$ ($F = 7.062$, $p = 0.000$). That is, the dependent variables, brand attitude, intention to purchase, attitude towards the company, attitude towards the CEO, and brand loyalty, showed significant differences at the 0.05 significance level. However, analysis of interactions showed insignificant results, with Wilks's $\lambda = 0.972$ ($F = 11.319$, $p = 0.249$). The dependent variables showed no statistically significant differences at the 0.05 significance level.

**Table 1.** Results of MANCOVA.

| Treatments | | Wilks's Lambda | F | df | p |
|---|---|---|---|---|---|
| Pre vs. After | | 0.928 | 3.496 ** | | 0.002 |
| Negative vs. positive | D.V | 0.951 | 2.329 ** | (1.275) | 0.033 |
| Types of CSR activity | | 0.864 | 7.062 *** | | 0.000 |
| Interaction | | 0.972 | 1.319 | | 0.249 |

Note: Interaction: Negative and positive x types of CSR activity; * $p < 0.1$, ** $p < 0.05$, *** $p < 0.01$.

As seen in Table 2, the study involved analyzing the average differences in attitude towards the brand, intention to purchase, image of the company, attitude toward the CEO, and brand loyalty in the group that was exposed to negative information about the company. First, the study looked at differences in the dependent variables before and after exposure to the negative information. Scores for attitude towards the brand decreased, on average, from 3.24 before exposure to negative information to 2.72 after exposure ($t = -5.515$, $p < 0.001$). Scores for intention to purchase, on average, also decreased, from 3.36 to 2.98 ($t = -4.560$, $p < 0.001$), as did scores for corporate image, from 2.98 to 2.55 ($t = -4.845$, $p < 0.001$). We found similar results for attitude towards the CEO (with scores decreasing on average from 2.57 to 2.12 ($t = -4.278$, $p < 0.001$)) and brand loyalty (with scores decreasing on average from 3.07 to 2.73 ($t = -4.604$, $p < 0.001$)).

Next, we analyzed average differences in scores for attitude towards the brand, purchase intention, corporate image, attitude towards the CEO, and brand loyalty in the group that was exposed to positive information about the company. In this case, scores for attitude towards the CEO and brand loyalty showed significant differences, on average. Scores for attitude towards the CEO increased, on average, from 2.63 to 2.92 ($t = 2.434$, $p = 0.018$). Scores for brand loyalty, on average, increased from 3.07 to 3.24, with marginal statistical significance ($t = 1.750$, $p = 0.085$, $p < 0.10$). However, scores for the remaining variables, including attitude towards the brand, intention to purchase, and corporate image showed insignificant differences on average. Thus, hypotheses 1-1 through 1-4 were supported. Test subjects' perceptions or intentions became more negative after exposure to negative information about the company, so hypotheses 2-1 and 2-2 were not supported. Hypotheses 2-3 and 2-4 were, however,

supported. The subjects' perceptions of the brand and attitudes toward the CEO became more positive after exposure to positive information about the company.

**Table 2.** Results of mean difference test for negative vs. positive information.

| Variable | Information | M(S.D) | Exposure | M(S.D) | Difference (S.E) | t | p |
|---|---|---|---|---|---|---|---|
| Brand | Negative | 2.98(0.723) | Before | 3.24(0.779) | −0.517(0.094) | −5.515 *** | 0.000 |
| | | | After | 2.72(0.888) | | | |
| | Positive | 3.24(0.732) | Before | 3.20(0.810) | 0.089(0.094) | 0.939 | 0.352 |
| | | | After | 3.29(0.829) | | | |
| PI | Negative | 3.17(0.681) | Before | 3.36(0.65) | −0.375(0.082) | −4.560 *** | 0.000 |
| | | | After | 2.98(0.782) | | | |
| | Positive | 3.45(0.696) | Before | 3.44(0.765) | 0.028(0.097) | 0.286 | 0.776 |
| | | | After | 3.47(0.817) | | | |
| Image | Negative | 2.76(0.644) | Before | 2.98(0.696) | −0.4290(0.089) | −4.845 *** | 0.000 |
| | | | After | 2.55(0.811) | | | |
| | Positive | 3.15(0.688) | Before | 3.08(0.720) | 0.139(0.087) | 1.579 | 0.120 |
| | | | After | 3.22(0.813) | | | |
| CEO | Negative | 2.34(0.712) | Before | 2.57(0.780) | −0.446(0.104) | −4.278 *** | 0.000 |
| | | | After | 2.12(0.916) | | | |
| | Positive | 2.78(0.641) | Before | 2.63(0.671) | 0.283(0.901) | 2.434 ** | 0.018 |
| | | | After | 2.92(0.882) | | | |
| Loyalty | Negative | 2.90(0.630) | Before | 3.07(0.683) | −0.346(0.075) | −4.604 *** | 0.000 |
| | | | After | 2.73(0.743) | | | |
| | Positive | 3.16(0.636) | Before | 3.07(0.700) | 0.167(0.095) | 1.750 * | 0.085 |
| | | | After | 3.24(0.769) | | | |

Note: Brand: Attitude toward the brand; PI: Purchase intention; Image: Corporate Image; CEO: Attitude toward CEO; Loyalty: Brand loyalty; * $p < 0.1$, ** $p < 0.05$, *** $p < 0.01$.

Regarding hypotheses 3 and 4, we then analyzed the average differences in the scores that subjects recorded for attitude towards the brand, purchase intention, corporate image, attitude towards the CEO, and brand loyalty in the group that was exposed to information about CSR activity that involved environmental preservation.

We first looked at differences in scores for the dependent variables before and after exposure to information about the environmental preservation CSR activity. As seen in Table 3, scores for attitude towards the brand on average increased, from 3.21 to 3.41, after exposure to the information regarding the CSR activity involving environmental preservation ($t = 1.760$, $p < 0.092$, $p < 0.10$). Scores for intention to purchase, on average, increased, from 3.44 to 3.71 ($t = 1.879$, $p < 0.002$). Scores for corporate image, attitude towards the CEO, and brand loyalty were statistically insignificant, although, on average, their values increased. Scores for corporate image increased, on average, from 3.18 to 3.33, those for attitude towards the CEO increased on average from 2.51 to 2.73, and those for brand loyalty increased, on average, from 3.25 to 3.26.

Next, the study analyzed average differences in scores for attitude towards the brand, purchase intention, corporate image, attitude towards the CEO, and brand loyalty in the group that was exposed to information about the CSR activity that involved social welfare support for developing countries. Scores for attitude towards the brand, on average, decreased from 3.18 to 2.91 ($t = -3.217$, $p < 0.002$) and scores for intention to purchase, on average, decreased significantly, from 3.33 to 3.14 ($t = -2.605$,

$p < 0.010$). Scores for corporate image, on average, decreased, from 2.96 to 2.75 ($t = -2.737$, $p < 0.007$), as did scores for brand loyalty, from 3.03 to 2.89 ($t = -2.160$, $p < 0.033$). However, scores for attitude towards the CEO showed, on average, only a statistically insignificant decrease, from 2.57 to 2.45 ($t = -1.258$, $p = 0.212$).

Thus, hypotheses 3-1 and 3-2 (which pertained to a scenario in which the company's CSR activity focused on preserving the environment) were supported, as subjects' scores for attitudes toward the brand and purchase intention increased following exposure to the CSR activity. On the other hand, hypotheses 3-3, 3-4, and 3-5 were not supported. Hypotheses 4-1, 4-2, 4-3, and 4-5 (which pertained to a scenario in which the company's CSR activity focused on contributing to social welfare by improving medical care in developing countries) were supported, but hypothesis 4-4 was not. Scores for attitude towards the brand, purchase intention, corporate image, and brand loyalty decreased following exposure to information about CSR activity that focused on supporting welfare in developing countries (see Tables 3 and 4, and Figures 1–5).

**Table 3.** Results of mean difference test for CSR activity exposure.

| Variable | Type | M(S.D) | Exposure | M(S.D) | Difference | t | p |
|---|---|---|---|---|---|---|---|
| Brand | Environment | 3.31(0.870) | Pre | 3.21(1.00) | 0.208 | 1.760 * | 0.092 |
| | | | After | 3.41(0.818) | | | |
| | Global | 3.05(0.701) | Pre | 3.18(0.073) | −0.267 | −3.217 ** | 0.002 |
| | | | After | 2.91(0.879) | | | |
| PI | Environment | 3.58(0.785) | Pre | 3.44(0.837) | 0.264 | 1.879 * | 0.073 |
| | | | After | 3.71(0.875) | | | |
| | Global | 3.23(0.669) | Pre | 3.33(0.725) | −0.190 | −2.605 ** | 0.010 |
| | | | After | 3.14(0.822) | | | |
| Image | Environment | 3.31(0.712) | Pre | 3.28(0.784) | −0.056 | −0.492 | 0.627 |
| | | | After | 3.33(0.742) | | | |
| | Global | 2.85(0.661) | Pre | 2.96(0.684) | −0.213 | −2.737 ** | 0.007 |
| | | | After | 2.75(0.868) | | | |
| CEO | Environment | 2.63(0.824) | Pre | 2.51(1.04) | 0.222 | 1.093 | 0.286 |
| | | | After | 2.73(0.868) | | | |
| | Global | 2.51(0.690) | Pre | 2.57(0.703) | −0.115 | −1.256 | 0.212 |
| | | | After | 2.45(0.971) | | | |
| Loyalty | Environment | 3.26(0.695) | Pre | 3.25(0.852) | 0.014 | 0.092 | 0.927 |
| | | | After | 3.26(0.715) | | | |
| | Global | 2.96(0.622) | Pre | 3.03(0.679) | −0.149 | −2.160 ** | 0.033 |
| | | | After | 2.89(0.769) | | | |

Note: Brand: Attitude toward the brand; PI: Purchase intention; Image: Corporate Image; CEO: Attitude toward CEO; Loyalty: Brand loyalty; * $p < 0.1$, ** $p < 0.05$, *** $p < 0.01$.

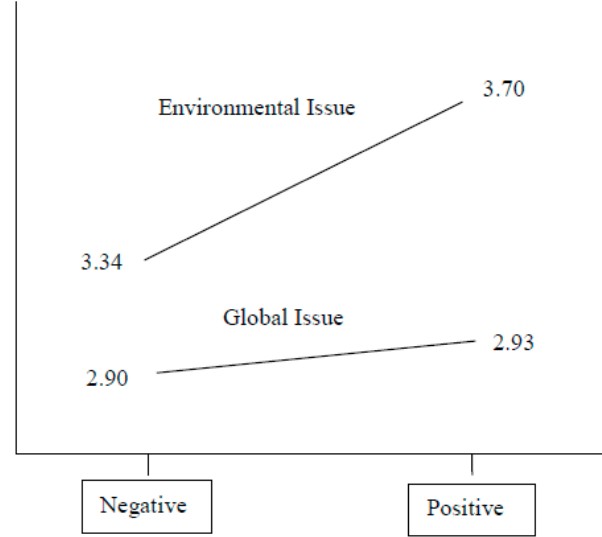

**Figure 1.** Attitude toward the Brand. Note: Environmental vs. Global Issues Prior to CSR Evaluation.

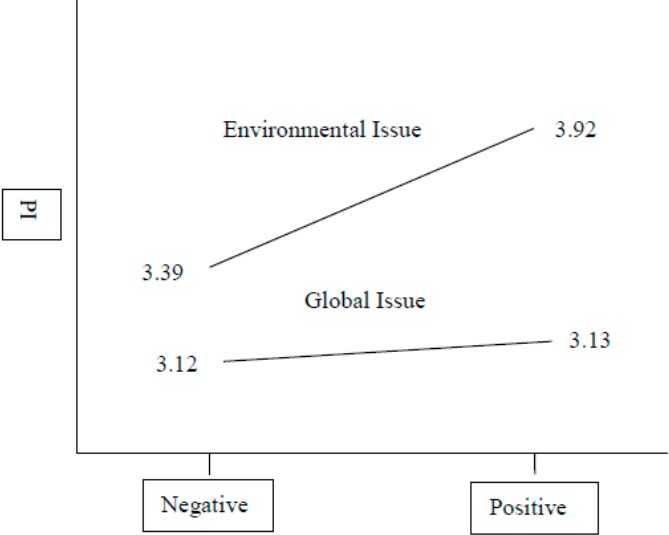

**Figure 2.** Purchase Intention. Note: Environmental vs. Global Issues Prior to CSR Evaluation.

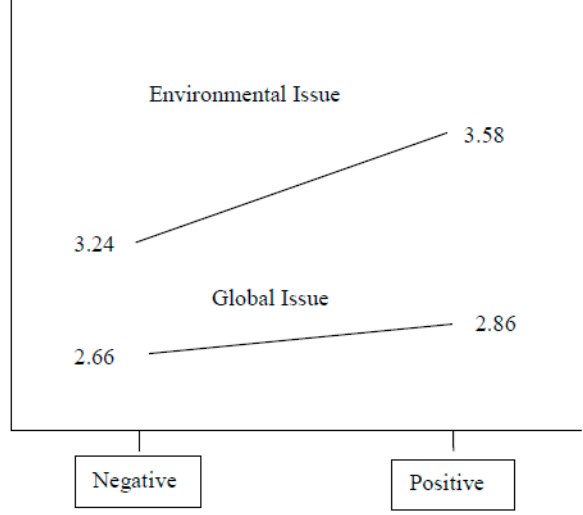

**Figure 3.** Corporate Image. Note: Environmental vs. Global Issues Prior to CSR Evaluation.

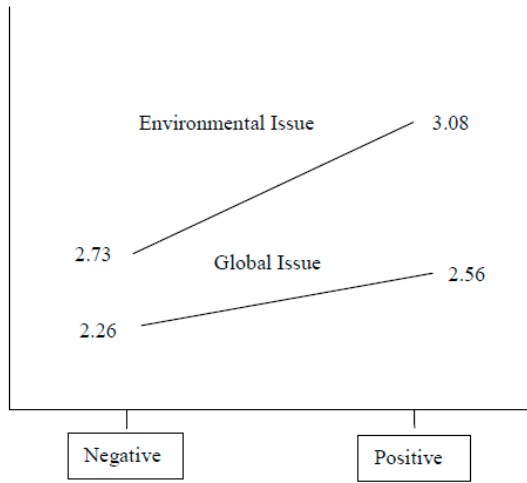

**Figure 4.** Attitude toward CEO. Note: Environmental vs. Global Issues Prior to CSR Evaluation.

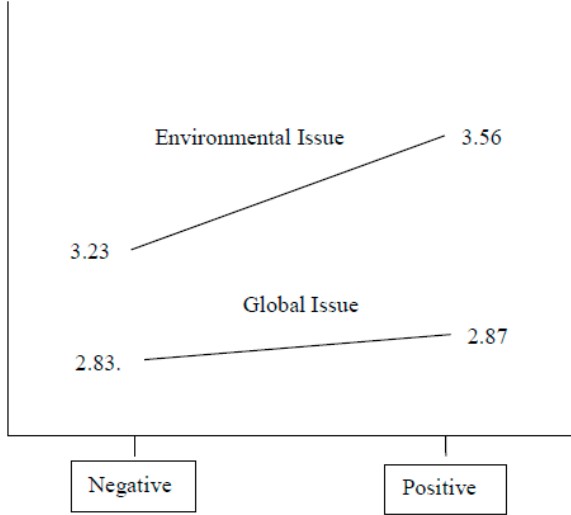

**Figure 5.** Brand Loyalty. Note: Environmental vs. Global Issues Prior to CSR Evaluation.

*5.2. Results for RQ 1*

The study explored how attitudes towards a company or brand differ before and after exposure to information about the company, or the company's CSR activity, and tested the role of the halo effect on these differences. The study's research questions focused on how exposure to information about a company or the company's CSR activity affects consumer attitudes towards the company or its brand and tested the extent to which any such effects were due to the halo effect. As reported in Table 4, the scores indicating the mean differences in ratings for each variable provided by test subjects provide evidence of the halo effect [14].

The study found that consumers' perceptions of or intentions towards the company or brand became more negative after exposure to negative information. Test subjects' attitudes toward the CEO and brand loyalty became slightly more positive after exposure to positive information. In particular, test subjects' brand attitudes became more positive and their purchase intentions strengthened following exposure to information that identified the company's CSR activity as focused on environmental preservation. However, the test subjects who were exposed to information indicating that the company's CSR activity focused on improving social welfare in developing countries (by improving medical care) exhibited more negative brand attitudes and purchase intentions, as well as a more negative brand image and weaker brand loyalty. These results indicate that there were mean differences before and after exposure to the stimuli.

**Table 4.** Result of halo effect.

| Pre-Exposure | | | D.V | After-Exposure | | | Mean difference | | | | Halo Effect | |
|---|---|---|---|---|---|---|---|---|---|---|---|---|
| N/P | Types | M(S.D) | | N/P | Types | M(S.D) | Differ | p | Differ | p | Differ | p |
| N | Envrio | 3.31(0.721) | Brand | N | Envrio | 2.83(0.800) | 0.48 | 0.002 | 1.05 | 0.000 | 0.84 | 0.001 |
| | Global | 3.14(0.854) | | | Global | 2.57(0.986) | 0.57 | 0.002 | | | | |
| P | Envrio | 3.41(0.747) | | P | Envrio | 3.54(0.740) | −0.13 | 0.034 | −0.21 | 0.006 | | |
| | Global | 3.14(0.824) | | | Global | 3.22(0.846) | −0.08 | 0.550 | | | | |
| N | Envrio | 3.42(0.809) | PI | N | Envrio | 2.99(0.730) | 0.43 | 0.002 | 0.76 | 0.000 | 0.68 | 0.001 |
| | Global | 3.31(0.730) | | | Global | 2.98(0.860) | 0.33 | 0.003 | | | | |
| P | Envrio | 3.47(0.751) | | P | Envrio | 3.49(0.827) | −0.02 | 0.774 | −0.08 | 0.550 | | |
| | Global | 3.33(0.839) | | | Global | 3.39(0.803) | −0.06 | 0.560 | | | | |
| N | Envrio | 3.06(0.678) | Image | N | Envrio | 2.70(0.756) | 0.36 | 0.003 | 0.88 | 0.000 | 0.61 | 0.001 |
| | Global | 2.87(0.751) | | | Global | 2.35(0.852) | 0.52 | 0.001 | | | | |
| P | Envrio | 3.23(0.699) | | P | Envrio | 3.36(0.738) | −0.13 | 0.037 | −0.27 | 0.003 | | |
| | Global | 3.04(0.727) | | | Global | 3.18(0.836) | −0.14 | 0.035 | | | | |
| N | Envrio | 2.62(0.747) | CEO | N | Envrio | 2.23(0.902) | 0.39 | 0.003 | 0.92 | 0.001 | 0.22 | 0.001 |
| | Global | 2.50(0.829) | | | Global | 1.97(0.926) | 0.53 | 0.001 | | | | |
| P | Envrio | 2.65(0.694) | | P | Envrio | 3.02(0.783) | −0.37 | 0.003 | −0.7 | 0.005 | | |
| | Global | 2.56(0.599) | | | Global | 2.89(0.854) | −0.33 | 0.003 | | | | |
| N | Envrio | 3.19(0.643) | Loyalty | N | Envrio | 2.99(0.659) | 0.2 | 0.006 | 0.61 | 0.001 | 0.13 | 0.038 |
| | Global | 2.92(0.716) | | | Global | 2.51(0.800) | 0.41 | 0.003 | | | | |
| P | Envrio | 3.11(0.709) | | P | Envrio | 3.33(0.666) | −0.22 | 0.005 | −0.48 | 0.001 | | |
| | Global | 2.95(0.678) | | | Global | 3.21(0.799) | −0.26 | 0.005 | | | | |

N: negative, P: positive, G: global support, E: environmental preservation, Brand: Attitude toward the brand; PI: Purchase intention; Image: Corporate Image; CEO: Attitude toward CEO; Loyalty: Brand loyalty; * $p < 0.1$, ** $p < 0.05$, *** $p < 0.01$, D.V: Dependent Variables, * $p < 0.1$, ** $p < 0.05$, ***$p < 0.01$.

A mean differences test showed that the mean values are statistically significant. The mean values of or scores, given the variables by subjects (for attitude towards the brand, purchase intention, corporate image, attitude towards the CEO, and brand loyalty), became more negative after subjects were presented with polarized information that described both the company's CSR activity and a scandal involving the company's CEO. Thus, the study confirmed that subjects' perceptions of or intentions regarding the company involved in the study differ before and after exposure to information about the company or the company's CSR activity, due to the halo effect.

## 6. Discussion and Conclusions

This study examines whether there is an interaction effect between the polarity of corporate information and two CSR activities and whether those CSR activities exerted a halo effect. The study involved an experiment designed to test whether subjects' attitudes changed with exposure to negative or positive information and whether their attitudes varied by type of CSR activity.

The results confirmed that there was a halo effect when responses were observed before and after exposure to information about the company that was targeted in the experiment. The results revealed that there was a mean difference in the responses given before and after subjects were exposed to information about the company's CSR activities. When presented with negative information, consumer attitudes are likely to become negative, whereas being exposed to positive information about a company's CSR activity (in particular, CSR activity related to environmental protection) leads to the formation of a positive attitude towards the company. Our findings suggest that the halo effect alters consumer attitudes towards a brand to a greater extent than it affects other variables, such as purchase intention, corporate image, attitude towards the CEO, and brand loyalty. That is, although the halo effect was present to some extent for all the variables included in the study, the effect was more powerful for perceptions of the brand than for the other variables.

The study's findings confirm that negative information about a company or brand triggers rational and emotional changes in consumers. Previous studies have reported that negative information not only provokes negative emotions, but also strongly affects thoughts and attitudes [73]. Therefore, consumer attitudes towards a brand or company become more negative when consumers are exposed to negative information. This critical finding suggests that when consumers are exposed to negative information about a company, their attitudes towards the brand, its CEO, and the company carrying the brand become more negative.

In this study's experiment, subjects' attention was directed to two types of CSR, one that focuses on environmental preservation, which was closely related to the company's image, and the other that focuses on medically related welfare projects in developing countries. The results indicate that the formation of an attitude or a change in an image was more positive when CSR activity was related to an established corporate image, which implies the relevance of the relationship between a company's image and type of CSR activity [74]. Significant interaction was observed only in the case of intention to purchase, implying that consumers' intention to purchase can be positively improved by the halo effect. There are, however, insignificant interaction results for brand attitude, attitude towards a company or CEO, and brand loyalty.

### 6.1. Theoretical Implications

A critical contribution of our study is the development of a theoretical framework for understanding the role of CSR activity in consumer attitude formation, as well as the causal relationship between CSR activity and consumer perceptions of a company before and after exposure to information about that activity. Some scholars have argued that consumers respond amicably and evaluate a company favorably if the company sponsors activities that are related to social issues [61]. Consumers evaluate a company or brand more amicably when CSR activity matches the image of the company or product well.

We analyzed whether providing information about CSR activity that is congruent with a corporate image generates a halo effect in the form of attitude changes. We found that the halo effect of CSR activity was much stronger when it involved protecting the environment than when it involved contributing to social welfare in developing countries, but such an effect depends, as noted above, on the brand image that consumers have formed about a company. CSR activity that is congruent with a company's image offers a more effective strategy for mitigating negative attitudes towards the brand or company or enhancing positive attitudes. This suggests that consumers perceive a company's image separately from such information and activity. Our findings suggest that a halo effect develops on the behavioral side, through image formation.

*6.2. Managerial Implications*

We derived two main managerial implications for managers seeking to develop effective CSR strategies. First, a company's stakeholders act as mediators who help consumers accept CSR activity that is highly congruent with a corporate image. Significant interaction was observed in subjects' attitudes towards the brand and company, implying that consumer perceptions and attitudes can be positively improved by the halo effect. When consumers realize that a company, about which they have been given negative information, had implemented CSR activities without comparing that information with their image of the company, they tend to form negative perceptions of the company. When CSR activity is closely related to a company or product, stakeholders can more easily promote consumers' association with the relevant company, which they develop using previously acquired knowledge or information. From a managerial standpoint, our findings should help marketers develop a CSR project with the potential to bring about attitudinal changes.

The findings further suggest that formulating a strategy that considers company and product image is critical for planning CSR activities. In general, CSR activity includes environmental protection, support for a local community, volunteer work, support for developing countries, consumer protection, social benevolence, and diversity. Selecting the right CSR activity from among many possibilities is a very important factor that can improve a company's image and enhance the benefits of its CSR activity. Pursuing CSR activities that are perceived to be too broad can harm the formation of a positive image about a company [23,73]. Although a company's philanthropy is often regarded as complementing CSR activity, too much emphasis on charity work can make consumers overestimate the philanthropic side of a company and negatively evaluate the CSR activity [74]. CSR activity certainly affects attitude formation or behavior, providing a competitive advantage over companies that do not implement CSR activity. CSR activity and the type of information about a company that consumers encounter is very important for improving established images of a company or brand [4,49].

Our research further highlights the importance of the effects of negative information on consumer attitude formation. The results of this study provide an opportunity to check the importance of negative information about a company or product, as well as the types of CSR activity that affect image formation. The study suggests that only CSR activities that are highly congruent with a company or its products can produce positive and amicable reactions from consumers through the halo effect. Information about a negative event provides a decisive clue that is soon followed by image deterioration. Changing images positively using the halo effect of CSR activity against such negative events should be considered when building corporate strategies.

Negative information about a company, such as reports of non-transparent management or poor product quality, accounting fraud, or a chief executive's ethical problems, exert a detrimental influence on the formation of brand images and attitudes, intention to purchase a product, and brand loyalty. The findings suggest that if a CEO's management practices are ethical or the CEO engages in activities that enhance global welfare, those practices and activities can strongly influence the formation of relevant consumer attitudes or intention to purchase, as well as the formation of attitudes towards a company or brand. Information related to a company therefore influences the formation of consumer images or attitudes. Once information that implies that a company or brand behaves

in what consumers perceive as a good way is stamped on consumers, it can be reflected in active consumer behavior.

Negative information, in the form of rumors and even reports of natural disasters, involves events that are not intended by a company and, hence, the company's responsibility is perceived to be relatively small. Meanwhile, events related to corruption reflect the intentions of the company and the company is therefore held responsible for the effects of corruption. In this case, the brand and corporate image loses credibility in the consumer's mind. Significant effects exist only when consumers recognize the high congruence between a CSR activity that a certain company implements and the motivation behind that activity. CSR activity, which is based on the concept that a company is concerned about social problems and participates in the process of solving them together with citizens, is important for maintaining smooth relationships with interested parties. This knowledge should prove helpful for marketing practitioners when they must craft CSR communication strategies for consumers.

The research informing this study makes significant contributions that should interest both academicians and practitioners, given that CSR and negative information effects are increasingly attracting attention in both worlds. For academicians who are interested in the relationship between CSR activities and negative information effects, as well as the halo effect, there is much scope for further research. Bringing to the fore the linkage between CSR, negative information, and the halo effect contributes to the theoretical understanding of the integration of consumer information processing and business practices. From this perspective, this study has contributed valuable findings pertaining to CSR strategy by focusing on the relationship between CSR activity and consumer attitudes toward a company and suggesting effective applications to corporate marketing strategy.

### 6.3. Limitations

The generalizability of the experimental stimulus may limit the study's applicability. The study chose exposure to negative and positive information about a company and brand as experimental stimuli. The study did not diversify types of information beyond this positive/negative binary. For example, neither dissatisfaction with a product or consumer service, nor any form of international disgrace, was considered in this study. Future studies will have to choose stimuli that expand the range of information types. This study divided CSR activities according to their congruence with a company or brand image, CSR activity that focuses on environmental preservation and fits a company's characteristics and image, and CSR activity focusing on welfare projects in developing countries. This study's findings might be limited insofar as the study did not specifically apply CSR activity types that were suggested by previous studies.

**Author Contributions:** J.-H.Y. wrote the paper and worked with L.-J.Y. to conceive and design the experiments L.-J.Y. and J.-H.Y. performed the experiments and analyzed the data; L.-J.Y. and J.-H.Y. contributed to parts of the experiments and the conclusions. Both authors made contributions to the work in this study.

**Funding:** This research received no external funding.

**Acknowledgments:** Thanks to the reviewers for their insightful comments in improving this paper and further highlighting the importance of this avenue of research.

**Conflicts of Interest:** The authors declare no conflict of interest.

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
