# Peer review of "The Halo Effect of CSR Activity: Types of CSR Activity and Negative Information Effects"

_sustainability, doi:10.3390/su11072067_

Round 1

Reviewer 1 Report

Dear Authors,

first of all congratulation for your research work. The manuscript is well-written and focuses on a much discussed topic among academics and marketers.

I would suggest only few adjustment:

- In lines 32-33 and 151-152 there is something wrong with text formatting;

- in lines 90-91 the word CSR is repeated twice;

- in lines 153-154 in the brackets should be the number of the reference instead of "Blum and Naylor, 1986".

Regards,

The reviewer

Author Response

Issue Number: sustainability-476261

Thank you to the reviewers for their insightful comments in improving this paper and further highlighting the importance of this avenue of research. Each point made by each reviewer has been addressed and a response is outlined below. The revised part is highlighted with red color.

Major comments and suggestions:

In comment 1, one reviewer asks the author to revise the test style and the missed reference in this paper.

Response: In response to this comment, the author enriched the style and references and then developed the whole text.

In comment 2, one reviewer asks the author to develop or add the main purpose of the study in introduction section.

Response: In response to this comment, the author develops the main purpose of the study on pages 3 through 4

In comment 3, one reviewer asks the author to revise the research question.

Response: In response to this comment, the author develops the research question in move after hypothesis section.

In comment 4, one reviewer asks the author to revise the results section in the mass of presented statistics. Author should be modified titles of Tables and Figures.

Response: In response to this comment, the author has revised the result section by removing repeated table on page 21 through 25. Author revised the titles of Tables and Figures.

Thank you for your wonderful comments, which guided my work in revising the paper.

Reviewer 2 Report

The paper presents an interesting topic of the halo effect of CSR activity. It produces some interesting and useful results. In general terms the paper is well written and well argued. However, some revisions are needed.

Research questions (section 1.1) are not formulated in accordance with the aims of the study.

There appears to be lack of consistency in the aims of the study as stated in the Introduction section of the manuscript and that stated under the section on Results.

Inconsistency is also apparent between RQs section and the Results section, (e.g. Why first put your attention to RG2, no to RG1?).

The quality of the details and differences tend to disappear in the mass of presented statistics.

Modify titles of Tables and Figures. You offer too long description, no titles.

Author Response

(The authors gave the same response as above.)
